# CAR Protects Females from Diet-Induced Steatosis and Associated Metabolic Disorders

**DOI:** 10.3390/cells12182218

**Published:** 2023-09-06

**Authors:** Fabiana Oliviero, Wendy Klement, Lucile Mary, Yannick Dauwe, Yannick Lippi, Claire Naylies, Véronique Gayrard, Nicola Marchi, Laila Mselli-Lakhal

**Affiliations:** 1Toxalim (Research Centre in Food Toxicology), Université de Toulouse, INRAE, ENVT, INP-Purpan, UPS, 31027 Toulouse, France; 2Institute of Functional Genomics, University of Montpellier, CNRS, INSERM, 34094 Montpellier, France

**Keywords:** NAFLD, sexual dimorphism, constitutive androstane receptor, high-fat diet

## Abstract

Non-Alcoholic Fatty Liver Disease (NAFLD) is the most common cause of chronic liver disease worldwide, affecting 70–90% of obese individuals. In humans, a lower NAFLD incidence is reported in pre-menopausal women, although the mechanisms affording this protection remain under-investigated. Here, we tested the hypothesis that the constitutive androstane nuclear receptor (CAR) plays a role in the pathogenesis of experimental NAFLD. Male and female wild-type (WT) and CAR knock-out (CAR−/−) mice were subjected to a high-fat diet (HFD) for 16 weeks. We examined the metabolic phenotype of mice through body weight follow-up, glucose tolerance tests, analysis of plasmatic metabolic markers, hepatic lipid accumulation, and hepatic transcriptome. Finally, we examined the potential impact of HFD and CAR deletion on specific brain regions, focusing on glial cells. HFD-induced weight gain and hepatic steatosis are more pronounced in WT males than females. CAR−/− females present a NASH-like hepatic transcriptomic signature suggesting a potential NAFLD to NASH transition. Transcriptomic correlation analysis highlighted a possible cross-talk between CAR and ERα receptors. The peripheral effects of CAR deletion in female mice were associated with astrogliosis in the hypothalamus. These findings prove that nuclear receptor CAR may be a potential mechanism entry-point and a therapeutic target for treating NAFLD/NASH.

## 1. Introduction

Metabolic diseases include many energy metabolism disorders, such as obesity and type II diabetes. Metabolic diseases represent a global health issue approaching epidemic proportions. Non-Alcoholic Fatty Liver Disease (NAFLD) emerges as the most common cause of chronic liver disease worldwide, affecting 30% of the general population and 70 to 90% of obese individuals [1]. NAFLD is characterized by a reversible accumulation of lipid droplets in the liver, which in some cases can progress to more severe states such as Non-Alcoholic Steato-Hepatitis (NASH), cirrhosis, or hepatocellular carcinoma. NAFLD is a multifactorial pathology resulting from genetic and/or environmental factors such as lack of physical exercise and high fat and carbohydrate diets. Moreover, several studies suggest a sexual dimorphism in the prevalence of NAFLD with a higher susceptibility in men compared to pre-menopausal women [2]. Despite the important prevalence of hepatic diseases, there is no approved treatment for NAFLD. Therefore, it is essential to elucidate the molecular mechanisms involved in the pathogenesis of NAFLD to seek new therapeutic targets while considering the sexual dimorphism of this disease. Nuclear receptor CAR (Constitutive Androstan Receptor), mainly expressed in the liver, has first been described for its role in exogenous detoxification processes and endogenous catabolism of steroid and thyroid hormones [3,4]. However, many studies have highlighted the involvement of CAR in major hepatic metabolic pathways such as gluconeogenesis and lipogenesis, thus revealing a role in regulating energy homeostasis [5,6]. Given its role at the crossroads between endocrine regulation and energy metabolism, CAR could be involved in dimorphic metabolic disruptions such as NAFLD. Using a CAR knock-out murine model, this study demonstrates that CAR contributes to protection against metabolic disorders induced by a HFD diet. This protection is more prominent in females, where CAR deletion resulted in more severe weight gain, hepatic steatosis, liver injury, hyperglycemia, hyperinsulinemia, and hypothalamic astrogliosis compared to WT females. Importantly, CAR deletion leads to dimorphic disruptions of the hepatic transcriptome, with an up-regulation of genes involved in a NAFLD to NASH transition, specifically in female mice. The transcriptomic analysis also highlighted a possible cross-talk between CAR and ERα receptors in female mice, potentially explaining the protective role of CAR.

## 2. Materials and Methods

### 2.1. Animal Experiments

All in vivo experiments were conducted following French national and European laws and regulations relating to the housing and use of animals. These experiments have been approved by an independent ethics committee (Toxcométhique, INRAE ToxAlim, Toulouse, France). CAR knock-out mice backcrossed on C57BL/6 J background were provided by Dr. Urs A. Meyer (Biocenter, University Basel, Switzerland). The 8-week-old male and female, wild-type (WT) or CAR knock-out (CAR−/−) mice were fed a high-fat diet (HFD, 60% fat, D12492, Research Diets, New Brunswick, NJ, USA) during 16 weeks (*n* = 10 mice per group in order to achieve statistical robustness, divided into two cages). Mice had access to food and water ad libitum with 12 h light/dark cycles (23+/−2 °C). Body weight, food, and water intake were measured weekly. Animals were euthanized following 16 weeks of HFD by cervical dislocation. Randomization was introduced during animal sacrifice. Liver, subcutaneous (SC WAT)) and perigonadal (PG WAT) white adipose tissues were collected, weighed, snap-frozen in liquid nitrogen, and stored at −80 °C for further use. The adiposity index (AI) was calculated as followed AI% = (SC WAT + PG WAT)/Body weight.

### 2.2. Oral Glucose Tolerance Tests

At 4 and 10 weeks of diet, oral glucose tolerance tests were performed on conscious mice. Mice were fasted for 6 h and fasted glycaemia was assessed with a drop of tail vein blood using an AccuCheck Performa glucometer (Roche Diagnostics, Indianapolis, IN, USA). Mice then received an oral glucose load (2 g/kg body weight) and blood glucose was assessed as described above at −15, 0, 15, 30, 45, 60, 90, and 120 min.

### 2.3. Liver Histological Sections and Scoring

Paraformaldehyde-fixed, paraffin-embedded liver tissue was sliced into 3-μm sections, deparaffinized, rehydrated, and stained with hematoxylin-eosin or Sirius red for histopathological analysis. Slides with two sections of a single hepatic lobe were digitized using a Mirax scanner from 3DHistech. Sections were visualized using CaseViewer software version 2.4.0.119028 and were individually scored using a NAFLD scoring system (NAS) adapted from Kleiner et al. [7]. ImageJ public domain software (ImageJ Website, https://imagej.net/ij/, accessed on 5 January 2023) was used to assess the area covered by lipid droplets or Sirius red staining. A total of four variables were qualitatively assessed and ranked with a score: hepatocellular steatosis, liver inflammation, lobular fibrosis. A detailed summary of the criteria for score assignments is presented in Appendix A.

### 2.4. Hepatic Neutral Lipids Analysis

Hepatic neutral lipid contents were determined at the end of the experiment as described previously [8]. Liver samples were homogenized in methanol/5 mM EGTA (2:1, *v*/*v*); lipids were extracted with chloroform/methanol/water (2.5:2.5:2.1, *v*/*v*/*v*), in the presence of glyceryl trinonadecanoate, stigmasterol, and cholesteryl heptadecanoate (Sigma Saint-Quentin Fallavier, France) as internal standards. Triglycerides, free cholesterol, and cholesterol esters were analyzed by gas-liquid chromatography on a Focus Thermo Electron system.

### 2.5. Plasma Analysis

Blood samples were collected following a 6 h fast for insulinemia assays or in fed-state before euthanasia for aspartate transaminase (AST) and alanine transaminase (ALT) assays. Blood was collected from the sub-mandibular vein using a lancet into lithium-heparin coated tubes (BD Microtainer). Plasma was obtained by centrifugation (1500× *g*, 10 min, 4 °C) and stored at −80 °C. Fasted insulinemia was assayed using the ultrasensitive mouse insulin ELISA kit (Crystal Chem, Elk grove Village, IL, USA). AST and ALT plasmatic levels were assayed using a PENTRA 400 biochemical analyzer (Anexplo facility, Toulouse, France).

### 2.6. Microarray and qPCR Gene Expression Studies

RNA was extracted from liver samples and qPCR assays were performed as previously described [9] for gene expression analysis of *Tnfα, Il1β*, *Pdgfr1β, Col1a1*, *Tgfb1*, *Tgfbr1*, *Acta2*. Primers used for qPCR assays are reported in Appendix A. Microarray analysis on liver samples (*n* = 6 per group) was performed at the GeT-TRiX facility (GénoToul, Génopole Toulouse) using Agilent Sureprint G3 Mouse GE v2 microarrays (8 × 60 K, design 074809) following the manufacturer’s instructions. Microarray data were analyzed using R (R Core Team, 2018) and Bioconductor packages [10]. Raw data (median signal intensity) were filtered, log2 transformed, corrected for batch effects (microarray washing bath and labelling serials) and normalized using the quantile method [11]. A model was fitted using the limma lmFit function [12]. Pair-wise comparisons between biological conditions were applied using specific contrasts. A correction for multiple testing was applied using the Benjamini–Hochberg procedure [13] to control the False Discovery Rate (FDR). Probes with FDR ≤ 0.05 were considered to be differentially expressed between conditions. Hierarchical clustering was applied to the samples and the differentially expressed probes using the 1-Pearson correlation coefficient as distance and Ward’s criterion for agglomeration. The clustering results are illustrated as a heatmap of expression signals. The enrichment of KEGG pathways was evaluated using Enrichr (Biotools). The differentially expressed gene list of CAR−/− females was in silico compared in BaseSpace Correlation Engine (Illumina, NextBio) to liver signatures of female ERα−/− knock-out mice fed an HFD (D12492, Research Diets) for 10 weeks (GSE95283).

### 2.7. Brain Immunohistochemistry

Brains were fixed in PFA 4% solution and immersed in sucrose 15% for 24 h followed by sucrose 30% (*n* = 5 per group). Brains were then snap-frozen and stored at −80 °C. Slices (20 μm) were obtained using a cryostat, and immunohistochemistry was performed after PBS washes. Slices were added with blocking solution (PBS, triton 0.5%, horse serum 20%) at room temperature for 1 h. Primary antibodies (Appendix A) were diluted in a blocking solution, and slices incubated overnight at 4 °C. After PBS washes, a secondary antibody was added in PBS for 2 h at room temperature. After PBS washes, slices were mounted using Vectashield containing DAPI. For all quantifications, 20X Z-stack images (Z = 12 to 15 planes, each of 1 µm) were analyzed using Fiji (version 2.9.0). Two slices were examined for each mouse to quantify signals in constant regions of interest (ROI), identified by DAPI maps: arcuate (AN) and paraventricular (PVN) nuclei of the hypothalamus, cortex (CTX), dentate gyrus (DG), and white matter (WM). Before analysis, all Z-stacks images were combined (Z-project, sum) using Fiji. For GFAP quantification: images were converted to RGB stack format. The signal threshold was adjusted to 200 units for each image. The area of GFAP signal was calculated, setting threshold sensitivity equal for each image. GFAP data are expressed as a percentage of ROI total pixels. For *Iba1* quantification, a cell counter tool was used to calculate the total number of DAPI cells and the number of *Iba1*+ cells in each ROI. Data are expressed as a percentage (*Iba1*+/tot DAPI+) × 100.

### 2.8. Data Representation and Statistical Analysis

All data are presented as the mean +/− standard error of the mean (SEM). Differential effects were analyzed on GraphPad Prism (version 9.00; GraphPad Software) to evaluate the effect of CAR deletion combined with high-fat diet in male or female mice. For each sex, the WT group was compared to the CAR−/− group using two-tailed Student’s *t*-test. To compare histological score ranks, the Mann–Whitney test was used. A *p*-value < 0.05 was considered significant.

## 3. Results

### 3.1. CAR Deletion Exacerbates HFD-Induced Body Weight Gain Only in Female Mice

A weekly follow-up revealed a higher body weight in CAR−/− than in WT males from week 2 until week 14 (Figure 1A). Male mice present a total weight gain of 25.7 g for WT and 26.2 g for CAR−/− (Figure 1A). In contrast, CAR−/− female mice gain 30.5 g compared to 14.5 g in WT (Figure 1B). At 16 weeks, the mean weight was 47.3 g for WT males and 49.5 g for CAR−/− males. CAR−/− females reached 49.2 g compared to 33.9 g in WT females. No difference was observed in the food and water intake of WT and CAR−/− mice (Appendix A). Next, the subcutaneous and perigonadal white adipose tissues (WAT) were harvested and weighed (Figure 1D). Subcutaneous WAT was 0.049 g in WT compared to 0.073 g in CAR−/− females. Perigonadal WAT was increased from 0.052 g in WT to 0.068 g in CAR−/− females. The adiposity index was calculated taking into account perigonadal and subcutaneous adipose tissue was greater in CAR−/− female mice compared to WT.

### 3.2. CAR Deletion Exacerbates HFD-Induced Hyperglycemia and Hyperinsulinemia

After 10 weeks of diet, glucose tolerance was assessed using an oral glucose tolerance test (OGTT). WT and CAR−/− males presented equal glucose tolerance as indicated by an equal OGTT area under the curve (AUC) (Figure 2A,C). Conversely, CAR−/− female mice present decreased AUC, revealing better glucose tolerance than WT mice (Figure 2B,C). Fasted glycaemia and insulinemia levels were assessed, revealing more important hyperglycemia and hyperinsulinemia in CAR−/− mice than WT in both male and female mice (Figure 2D,E).

### 3.3. CAR Deletion Exacerbates HFD-Induced Hepatic Steatosis and Liver Injury

After 16 weeks of HFD, liver samples were harvested for histological sectioning, hematoxylin-eosin staining, and scoring of hepatocellular steatosis. Analysis of the area covered by the lipid droplets showed that the steatosis was increased two times more in CAR−/− females (↑19.12%) than in CAR−/− males (↑10.29%). WT males presented mainly microvesicular hepatocellular steatosis (Figure 3A), with a mean steatosis score of 3.33. CAR−/− males showed a higher mean steatosis score of 5.00, with cells presenting mixed micro/macrovesicular or microvesicular steatosis (Figure 3A). In WT females, a score 2 microvesicular steatosis was observed in all animals, which is considerably lower than that of WT males (Figure 3A). CAR−/− females presents a more severe steatosis, with a mean score of 4.80, comparable to that described for CAR−/− males (Figure 3A).

Hepatic neutral lipids were analyzed to support the histological observations. CAR−/− males present increased levels of cholesterol esters compared to WT mice (Figure 3B). In females, both cholesterol esters and triglycerides increase in CAR−/− mice (Figure 3B).

Subsequently, we determine whether the observed steatosis presents progression markers toward a more severe form, such as NASH. The presence of foci of mononuclear and polymorphonuclear cells assessed inflammation scores. CAR−/− male mice presented a discrete score-1 inflammation with changes comparable in appearance and distribution to WT mice (Figure 3C). Score-1 inflammation was also observed in WT and CAR−/− females (Figure 3C). Despite no morphological differences, RNA expression of pro-inflammatory cytokine *Tnfα* and *Il1β* was increased in CAR−/− mice (Figure 3C).

To further characterize the hepatic impact of CAR deletion, plasmatic levels of alanine aminotransferase (ALT) and aspartate aminotransferase (AST) were assessed to evaluate liver injury. Both CAR−/− males and females, presented increased levels of ALT and AST, suggesting underlying steatohepatitis (Figure 3D).

The presence of liver fibrosis was assessed by analysis of histological slices stained with Sirius red (Figure 4A). The analysis of fibrosis area coverage demonstrated no variation between CAR−/− and WT mice, irrespective of their gender (Figure 4B). Fibrosis scoring revealed a comparable score-1 peri-sinusoidal or peri-portal fibrosis in all groups and no transcriptional changes in the expression of the *Pdgfr1β*, *Tgfb1*, *Tgfbr1*, and *Acta2* markers of fibrosis (Figure 4B). Only the expression of the marker Col1a1 was increased in females (Figure 4B).

### 3.4. Dimorphic Impact of CAR Deletion on Hepatic Transcriptome

Next, a microarray analysis was performed to assess the impact of CAR deletion on the hepatic transcriptome of HFD-fed mice. A principal component analysis (PCA) revealed an important effect of CAR deletion on the hepatic transcriptome in males and females (Figure 5A), with significant discrimination of WT and CAR−/− mice on the first principal component, representing 28.6% of the total variance (Dim1). Clustering of male and female mice is reported on the second principal component (Dim2, 22.7% of total variance) with a convergence of CAR−/− females towards the male profile.

A heat map of 3354 probes selected as differentially expressed genes was plotted (Figure 5B). Nine clusters are distinguishable; cluster 1 is down-regulated explicitly in CAR−/− males, whereas cluster 5 is specifically up-regulated in CAR−/− females. Clusters 4 and 6 are impacted by sex, whereas clusters 2, 8, and 9 are by genotype. The expression profiles of clusters 3 and 7 in CAR−/− female mice are similar to those of males. Enrichment analysis of Kegg pathways was performed for clusters 3 and 7. The 7 most significant pathways are presented in Appendix A.

Venn diagrams were plotted to compare variations between WT and CAR−/− in both sexes (Figure 5C,D). CAR−/− males and females present 1643 and 1896 up-regulated genes compared to WT. Only 810 up-regulated genes are common to CAR−/− males and females. Similarly, CAR−/− males and females present 1550 and 1788 down-regulated genes compared to WT, and only 367 down-regulated genes are common to both sexes. Thus, most up and down-regulated genes in CAR−/− mice are not common to both sexes, revealing a dimorphic impact of CAR deletion on the hepatic transcriptome. Enrichment analysis of up-regulated male genes revealed pathways such as osteoclast differentiation and natural killer cytotoxicity (Figure 5C). In CAR −/− females, genes involved in the NAFLD and thermogenesis pathways are up-regulated (Figure 5C). In males, steroid hormone biosynthesis and amino acid metabolism are negatively affected by CAR deletion (Figure 5D). Protein processing in the endoplasmic reticulum (ER) and spliceosome are the main pathways represented among down-regulated female genes (Figure 5D). The 36 genes linked to the NAFLD pathway, represented among the specifically up-regulated genes in CAR−/− females (Figure 5C), are in Figure 5E. These genes were traced in the NAFLD *Kegg pathway* map (hsa04932), representative of a stage-dependent progression of NAFLD, and are mainly involved in β-oxidation (*Ndufa-b-s-c*, *Sdha-b*, *Cyc1*, *Uqcrq-c-s*, *Cox1-5-6-7-8*) and inflammation (*Jun*) which are mechanisms disrupted in NASH conditions. Genes involved in simple steatosis mechanisms are specifically up-regulated genes in CAR−/− females (*Gsk3*, *Prkab2*).

Positive correlation between transcriptomic signatures of CAR−/− and ERα−/− females. The hepatic transcriptomic signature of CAR−/− males and females was compared to other hepatic profiles using *Base Space Correlation Engine (Illumina).* The transcriptomic signature of CAR−/− females positively correlated with ERα knock-out females fed an HFD (Figure 6A). Of up-regulated genes, 373 are common to CAR−/− females and ERα females (Figure 6B).

Enrichment analysis revealed that these genes are involved in NAFLD, and oxidative phosphorylation is disrupted in NASH conditions (Figure 6D). On the other hand, 345 down-regulated genes are common to CAR−/− and ERα females (Figure 6C) and are primarily involved in the spliceosome pathway and linoleic acid metabolism (Figure 6E).

### 3.5. CAR Deletion Associates with Hypothalamic Astrogliosis in HFD-Fed Females: Initial Evidence

We have previously demonstrated that CAR deletion leads to recognition memory impairment and increased anxiety-like behavior in males [14]. In this study, our aim was to investigate the response of the mouse brains to a HFD in order to understand how the combination of the CAR deletion and a HFD influences brain function. Specific brain regions were analyzed to assess the possible impact of CAR deletion and HFD on central nervous system homeostasis, initially exploring glial cells at a specific brain border. GFAP reactivity in the arcuate (AN) and paraventricular (PVN) nuclei of the hypothalamus was increased in CAR−/− females (Figure 7A,B).

In these specific conditions, astrogliosis is specific to the hypothalamus. Although limited to the quantification of fluorescence in the total tissue, GFAP immunoreactivity was not observed in other regions such as the Cortex (CTX), hippocampus (e.g., dentate gyrus), and the white matter (WM) (Figure 7C). Total GFAP expression was unchanged in males (Appendix A). With the method used, tissue IBA1 microglial reactivity was unchanged across experimental conditions except for the WM in males (Appendix A) and AN in females (Appendix A). These results suggest regionally limited pro-inflammatory changes, indicating a possible and specific involvement of hypothalamic astrocytes in female mice.

## 4. Discussion

The central focus of this study was to investigate the gender-specific variations in the function of the nuclear receptor CAR in response to HFD, a known trigger of metabolic stress. A prevailing trend in animal models is that males tend to be more prone to developing obesity, insulin resistance, hyperglycemia, and steatosis compared to females upon exposure to dietary challenges [15]. The outcomes of this study highlight the critical role of the nuclear receptor CAR in safeguarding females against the onset of metabolic disorders.

These revelations shed light on a more pronounced detrimental impact of CAR deficiency in females in response to an HFD. These effects encompass weight gain, adiposity, steatosis development, and hypothalamic astrogliosis. Conversely, the implications of CAR absence in males are predominantly linked to hyperglycemia, hyperinsulinemia, and hepatic injury. Prior research, predominantly conducted in males, has showcased that the activation of the CAR receptor through pharmacological agonists can ameliorate glucose and insulin tolerance and alleviate hepatic steatosis in animals afflicted with metabolic disorders [6,16]. Our present study extends this understanding by unveiling a more robust influence of the CAR receptor in females relative to males when exposed to an HFD.

An unexpected result concerns the enhanced glucose tolerance observed in CAR−/− females compared to control even though they exhibited elevated levels of blood glucose and insulin. This improved glucose tolerance was already observed in a previous study, where we explored the role of nuclear receptor CAR under standard dietary conditions through the characterization of both WT and CAR−/− male and female mice [9]. The underlying mechanisms driving this improved glucose tolerance remain complex, potentially involving intricate physiological interactions such as heightened glucose uptake or increased insulin sensitivity in distinct tissues such as the liver, muscles, or adipose tissue [17]. A more targeted investigation utilizing mice with CAR inactivation in specific tissues could provide a deeper comprehension of this intriguing observation.

The intricate involvement of CAR in the development of NAFLD and NASH has been a subject of conflicting findings in existing research. Activation of CAR has been associated with both beneficial effects, such as mitigating hepatic steatosis, and adverse outcomes, such as exacerbating hepatic fibrosis and hepatocarcinogenesis, depending on the experimental context [18,19]. Most of these investigations have been centered around males, leaving a gap in understanding CAR’s dimorphic impact. Our study contributes to filling this gap by revealing CAR’s pivotal role in safeguarding female mice against NAFLD in the context of an HFD. These findings align with broader research highlighting the inherent protection observed in pre-menopausal women, which tends to diminish following menopause. Consistent with our results, a recent meta-analysis involving 54 studies reported a 19% lower risk of NAFLD in women compared to men [20].

The intricate connection between metabolism and the central nervous system has garnered increasing attention due to its potential to influence various physiological processes. Our study takes a comprehensive approach by examining sex and region-specific adaptations within the brains of HFD-fed CAR−/− mice in comparison to their WT counterparts. This approach was guided by our earlier research, which indicated that CAR deletion is associated with notable adaptations in recognition memory and anxiety-like behavior, along with observable histological changes in glial cells [14]. This endeavor is particularly relevant within the context of a HFD, known to exert substantial effects on both metabolism and neural function [21]. Through this study, we specifically focus on histological brain markers that indicate modifications in astrocytes, with a particular emphasis on their response to an HFD. The significance of our findings is underscored by the identification of distinct histological changes in astrocytes within the hypothalamic paraventricular and arcuate nuclei of female CAR−/− mice exposed to a high-fat diet. In the hypothalamus, astrocytes perform various functions that can directly affect energy homeostasis, such as nutrient sensing and transport [22]. In addition, HFD-induced metabolic stress induces astrogliosis, described as a protective homeostatic response that restrains food intake in response to the diet [23]. The alterations observed in astrocytes suggest a potential link between CAR, neural adaptations, and the intricate regulation of metabolic processes. In essence, our study offers a novel perspective by bridging the gap between CAR’s known role in metabolic regulation and its potential influence on neural mechanisms. This exploration sheds light on the complex interplay between CAR, the central nervous system, and the metabolic responses observed in females exposed to a high-fat diet. By investigating these interactions, we contribute to a more holistic understanding of how CAR exerts its effects across physiological domains, which could have broader implications for addressing metabolic disorders in a comprehensive manner.

To deepen our comprehension of the underlying mechanisms responsible for CAR’s protective role, particularly in females, we executed an in-depth analysis of hepatic transcriptomes in both WT and CAR−/− mice subjected to an HFD. The comparison of transcriptomes between HFD-fed CAR−/− males and females revealed a noteworthy convergence, as indicated by both the principal component analysis (PCA) in Figure 5A and the heatmap representation in Figure 5B. Strikingly, the absence of CAR in females resulted in a transcriptomic profile more closely aligned with that of males. Notably, the impact of CAR deletion appeared to exert a more pronounced effect on females than on males, evident from the divergent numbers of up and down-regulated genes (Figure 5C,D), which align with the more severe metabolic disorders observed in CAR−/− females (Figure 1B and Figure 2B). The phenomenon of female liver masculinization resulting from CAR expression loss has been previously documented in various studies. This effect is attributed to the alteration of the 6α/15α-OH testosterone ratio, a recognized biomarker associated with liver masculinization [24]. This transition towards a more masculine liver pattern may involve the Stat5b pathway, which is acknowledged for its role in regulating liver masculinization or feminization [25] and its potential interaction with CAR [9]. So, everything happens as if the invalidation of CAR causes females to lose their protection against metabolic disorders, rendering them equally susceptible as males.

Specifically, clusters 3 and 7, highlighted in the heatmap, demonstrated a distinct disruption in CAR−/− females (Figure 5B). Analysis of these clusters underscored the trend of the female transcriptome drawing closer to the male profile in the absence of CAR. Remarkably, among the array of insights, 57 genes associated with thermogenesis exhibited specific up-regulation in CAR−/− female mice (Figure 5C). This intriguing finding suggests a potential perturbation in thermogenesis that could contribute to the observed metabolic phenotype in CAR−/− females particularly the improved glucose tolerance [26]. Unraveling the mechanisms behind this potential perturbation in thermogenesis not only sheds light on the intriguing biology of CAR−/− females but also presents a promising avenue for understanding how thermogenesis and glucose tolerance are interconnected in the broader context of metabolic health

Moreover, upon a more comprehensive analysis, a notable enrichment of genes associated with the NAFLD pathway was observed among the up-regulated genes, with a particularly pronounced effect in CAR−/− females (Figure 5C). These identified genes intricately align with the well-established NAFLD Kegg pathway map, effectively delineating a cascade from the initial stages of hepatic lipid accumulation (NAFLD) to the more advanced state of non-alcoholic steatohepatitis (NASH). This pathway involves multifaceted biological processes, encompassing inflammation, oxidative phosphorylation, and perturbations in lipid metabolism. This outcome is in line with the evaluation of gene expression related to fibrosis, which reveals an elevated expression of the Col1a1 gene associated with collagen synthesis, particularly evident in females (Figure 4C) even if the Sirius red staining of liver slices did not reveal changes in fibrosis levels between WT and CAR−/− mice. All this underscores the potential significance of the CAR receptor in influencing the critical metabolic NAFLD-to-NASH cascade processes. This provides an intriguing avenue for future investigations into the intricate interrelationships between CAR, hepatic fibrosis, and the dynamic shifts occurring in NAFLD progression, particularly in the context of gender disparities. The cumulative evidence underscores the potential substantive role played by the CAR receptor in steering the pivotal and intricate metabolic processes that orchestrate the progression from NAFLD to NASH. Consistent with our results, a recent meta-analysis involving 54 studies reported a 19% lower risk of NAFLD in women compared to men. However, once NAFLD is established, women share a similar risk of advancing to NASH, along with a 37% higher risk of advanced fibrosis [20].

Using the Base Space Correlation Engine analysis, we evidenced a positive correlation between HFD-fed CAR−/− mice and HFD-fed ERα−/− mice (Figure 6A). This correlation provides a potential explanation for the heightened severity of steatosis observed in CAR−/− HFD-fed female mice compared to the control group (Figure 3A,B). Intriguingly, this correlation sheds light on the co-regulation of 373 genes being up-regulated and 345 genes being down-regulated in the absence of either CAR or Erα. Remarkably, these co-regulated genes play integral roles in the previously described metabolic pathways of thermogenesis and NAFLD (Figure 6D). This insight reveals a lack of compensatory regulation between these two nuclear receptors, emphasizing their individual indispensability. This phenomenon may arise from shared response elements on the promoters of the aforementioned genes, implying a direct mechanism. Moreover, the involvement of CAR in estrogen catabolism [4], prompts the proposal of an indirect mechanism, suggesting that CAR potentially modulates ERα activity by regulating estrogen levels. This unexplored interplay between the CAR and ERα signaling pathways could imply estrogen inefficiency in the absence of CAR, similar to the absence of ERα [27]. It is noteworthy that the protection against NAFLD in pre-menopausal women is attributed to estrogens, which curtail hepatic lipid accumulation and dampen liver inflammation and fibrosis [28]. Supporting this notion, the silencing of ERα expression in the liver using adenoviral short hairpin RNA significantly amplifies hepatic triglyceride accumulation in HFD-fed C57BL/6 female mice [29]. Similarly, the deletion of ERα in the liver abolishes the protective effects of E2 against HFD-induced steatosis [27]. In our study, intriguing parallels emerge as certain functions of CAR intersect with those of ERα. The deletion of CAR in female mice accentuates the accumulation of lipids in the liver in response to a HFD diet, as illustrated in Figure 3B. This cross-talk between CAR and ERα could explain why females lose protection against metabolic disorders upon losing CAR and become as susceptible as males (Figure 5A).

## 5. Conclusions

In conclusion, this study provides valuable insights into the dimorphic pathogenesis of NAFLD, unveiling the protective role of the nuclear receptor CAR in mitigating HFD-induced metabolic disruptions, particularly in females. CAR emerges as a potential therapeutic target for addressing NAFLD/NASH, warranting further investigation. The multifaceted roles of CAR, from metabolic regulation to potential neural influences, present a comprehensive perspective on its influence on physiological processes, opening avenues for a holistic approach to managing metabolic disorders.

## Figures and Tables

**Figure 1 cells-12-02218-f001:**
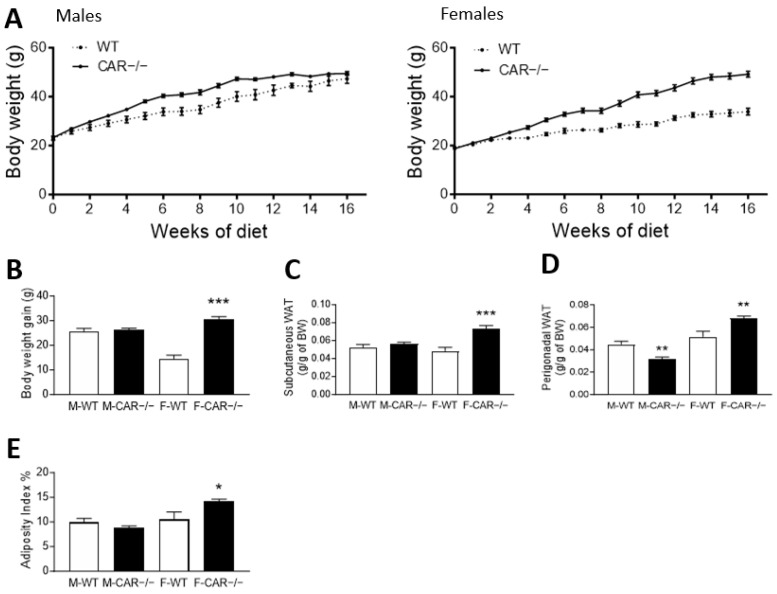
CAR deletion exacerbates HFD-induced body weight gain only in female mice. Body weight was monitored during 16 weeks of HFD in WT and CAR−/− male and female (**A**) mice and mean body weight gain was assessed (**B**). Following 16 weeks of diet, subcutaneous (**C**) and perigonadal (**D**) white adipose tissues (WAT) were harvested and weighed (BW: averaged by grams of body weight). The adiposity index was calculated taking into account perigonadal and subcutaneous adipose tissue (**E**) Data are the mean ± SEM of *n* = 10 per group. Groups were compared using two-tailed Student’s *t*-test and * *p* ≤ 0.05, ** *p* ≤ 0.01, and *** *p* ≤ 0.001.

**Figure 2 cells-12-02218-f002:**
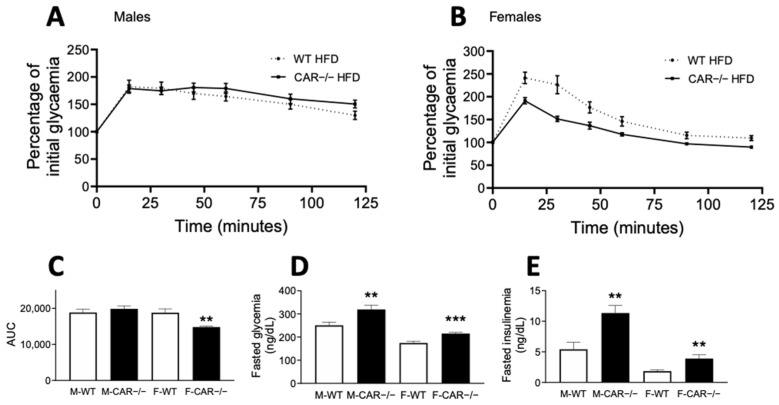
CAR deletion exacerbates HFD-induced hyperglycemia and hyperinsulinemia. At week 10 of HFD, an oral glucose tolerance test was performed (**A**,**B**). Data are presented as the percentage of initial glycaemia with the corresponding area under the curve (AUC, (**C**)). Glycaemia and insulinemia levels were assessed in the fasted state (**D**,**E**). Data are the mean ± SEM of *n* = 10 per group. Groups were compared using two-tailed Student’s *t*-test and ** *p* ≤ 0.01, and *** *p* ≤ 0.001.

**Figure 3 cells-12-02218-f003:**
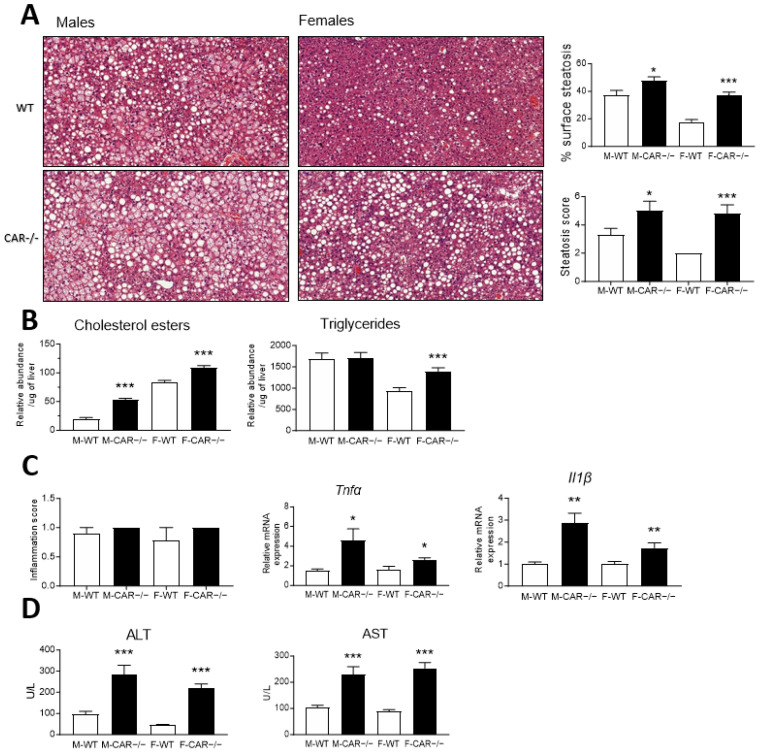
CAR deletion exacerbates HFD-induced hepatic steatosis and injury. Following 16 weeks of HFD, histological sections of the liver were stained with hematoxylin-eosin (HE, magnification ×20), the area covered by the lipid droplets was estimated and a score ranging in severity from 1 to 3 was assigned for hepatocellular steatosis (**A**). Hepatic cholesterol esters and triglycerides were analyzed by gas chromatography (**B**). Scoring of inflammation was performed on HE slices, and gene expression of inflammation markers Tnfα and Il1β were assessed by qPCR (**C**). Plasmatic levels of alanine aminotransferase (ALT) and aspartate aminotransferase (AST) were assayed (**D**). Data are the mean ± SEM of *n* = 10 per group. Groups were compared using two-tailed Student’s *t*-test and * *p* ≤ 0.05, ** *p* ≤ 0.01, and *** *p* ≤ 0.001. Scoring ranks were compared using the Mann–Whitney test.

**Figure 4 cells-12-02218-f004:**
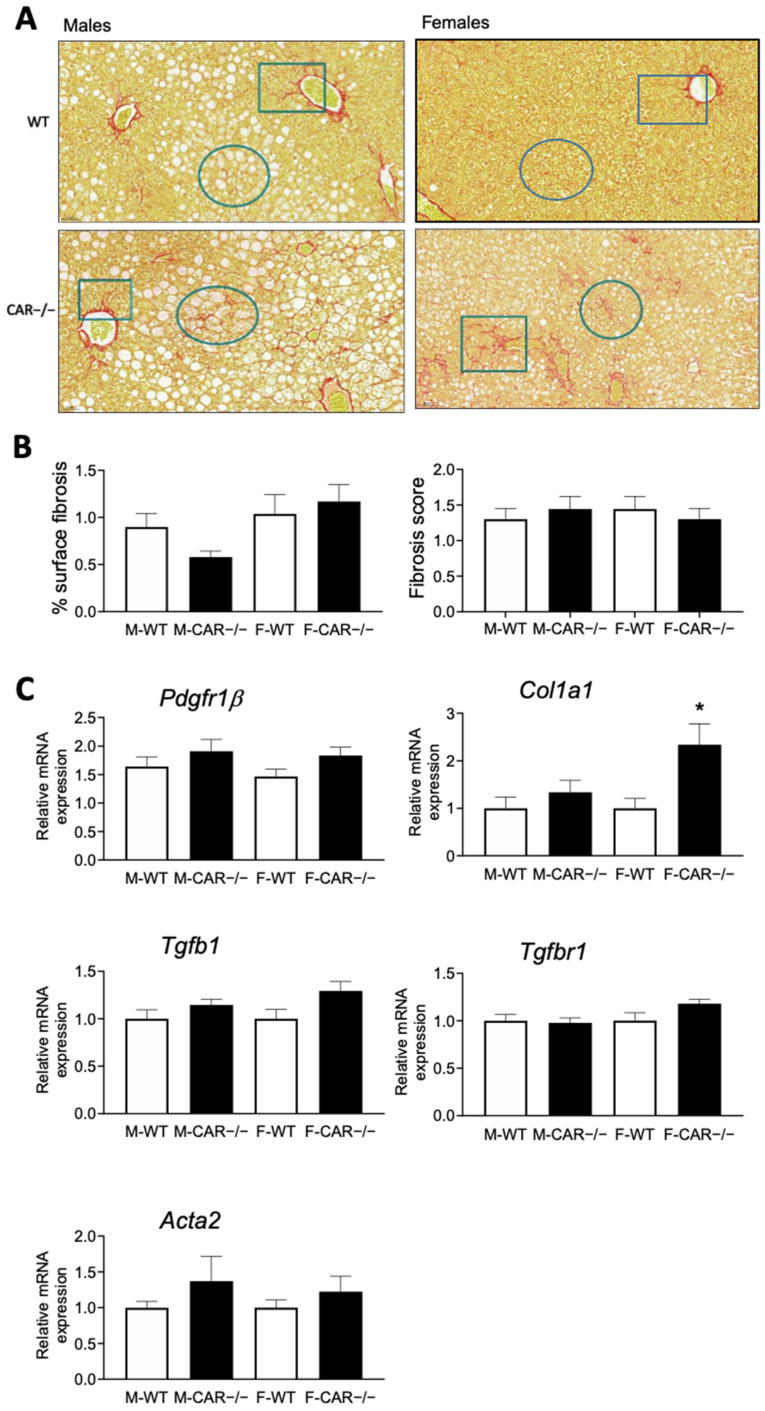
CAR deletion does not affect liver fibrosis. Histological sections of the liver were stained with Sirius red ((**A**), magnification ×20). Circles indicate peri-sinusoidal fibrosis, and rectangles indicate peri-portal fibrosis. Analysis of fibrosis area coverage was assessed, lobular fibrosis was scored on Sirius red slices and gene expression of fibrosis markers *Pdgfr1β, Col1a1*, *Tgfb1*, *Tgfbr1*, *Acta2* was assessed by qPCR (**B**,**C**). Data are the mean ± SEM of *n* = 10 per group. Groups were compared using two-tailed Student’s *t*-test and * *p* ≤ 0.05. Scoring ranks were compared using the Mann–Whitney test.

**Figure 5 cells-12-02218-f005:**
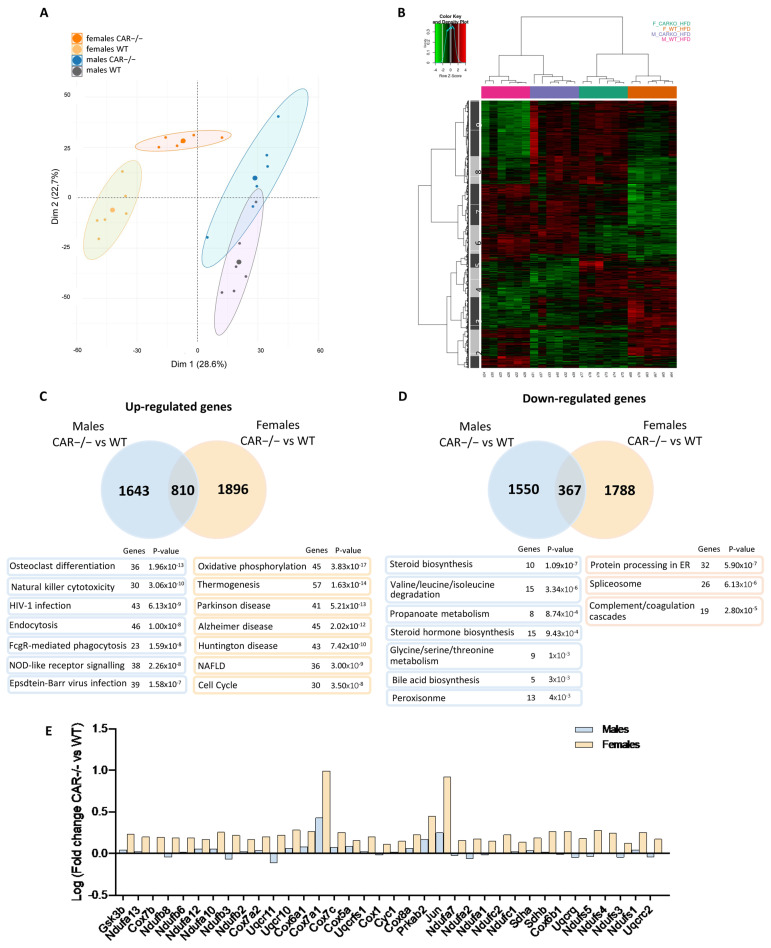
Dimorphic impact of CAR deletion on the hepatic transcriptome. Microarray analysis of hepatic transcriptome was performed. (**A**) Principal component analysis (PCA) with separation of data by two dimensions. (**B**) Heatmap representation of gene expression for each individual; the hierarchical clustering was obtained using Ward’s criterion and Pearson’s correlation coefficient. Red and green, respectively, indicate values above and below the mean averaged, centered and scaled expression values (Z-score). Black shows values close to the mean. Venn diagrams with the number of up-regulated (**C**) or down-regulated (**D**) genes in CAR−/− mice. Sex-specific variations were further analyzed, and enrichment of Kegg pathways is reported with gene number and corresponding *p*-value. (**E**) The 36 genes of NAFLD pathway which are specifically up-regulated in CAR−/− females were represented for both male and females.

**Figure 6 cells-12-02218-f006:**
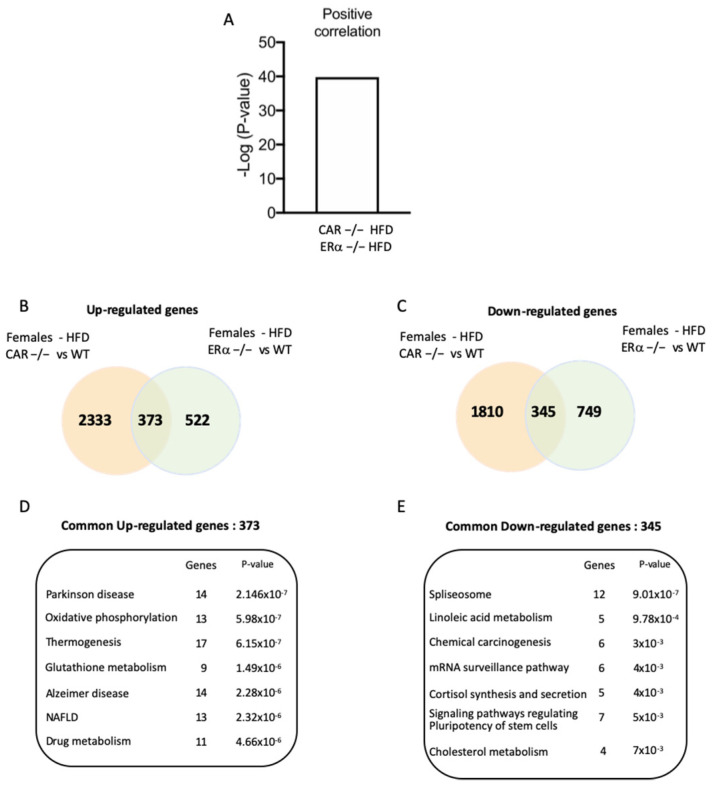
Positive correlation between transcriptomic signatures of CAR−/− and ERα−/− females. (**A**) The differentially expressed gene list of CAR−/− female mice was compared in Base Space Correlation Engine (Illumina) to publicly available datasets. The transcriptomic signature of CAR−/− females positively correlates with a significant *p*-value of overlap (−log(p-value overlap) > −log(0.05)) with the profile of ERα−/− females fed a HFD (GSE95283). Venn diagrams representing common up-regulated (**B**) and down-regulated genes (**C**) between CAR−/− and ERα−/− females. Common genes were further analyzed for enriched Kegg pathways using Enrichr (BioTools) (**D**,**E**).

**Figure 7 cells-12-02218-f007:**
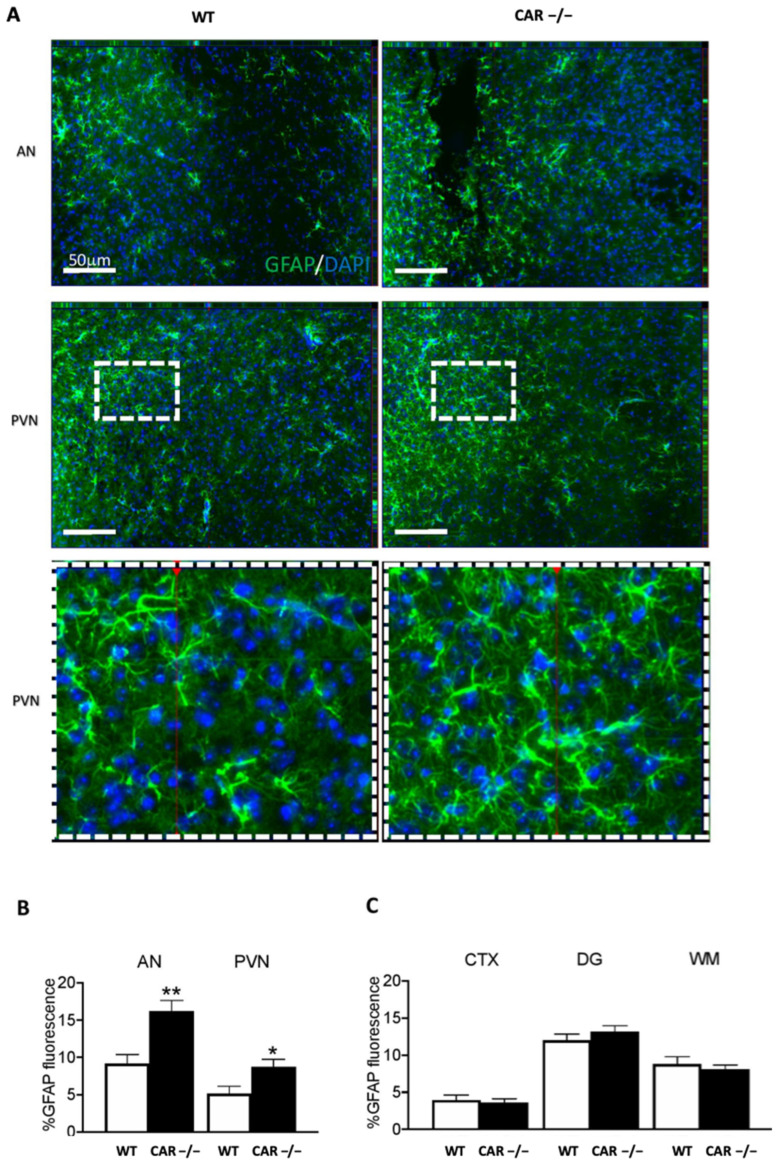
CAR deletion associates with hypothalamic astrogliosis in HFD-fed females. (**A**) Histological brain slices of WT and CAR−/− females were stained for GFAP expression, an astrocyte marker. Examples from the arcuate (AN) and paraventricular (PVN) nuclei of the hypothalamus. The white squares correspond to the regions for which a strong magnification is shown below. Quantification of GFAP fluorescence in AN and PVN (**B**), cortex (CTX), dentate gyrus (DG), and white matter (WM) of females (**C**). Data are the mean ± SEM of *n* = 5 per group. * *p* ≤ 0.05, ** *p* ≤ 0.01 using unpaired two-tailed Student’s *t*-test.

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
