# Peer review of "CAR Protects Females from Diet-Induced Steatosis and Associated Metabolic Disorders"

_cells, 2023, doi:10.3390/cells12182218_

Round 1

Reviewer 1 Report (Previous Reviewer 1)

Although authors still used an "ad populum" argument in order to justify the absence of mesenteric and retroperitoneal adipose tissue sampling, the manuscript improved. 

Author Response

We regret not having been able to collect the mesenterice and retroperitoneal adipose tissue to respond to the reviewer's request and will take into account this advice for our future experiments.

Reviewer 2 Report (New Reviewer)

In this study, the authors investigate the role of CAR in NAFLD progress and found that in CAR knock out mice model, CAR deficient results in mote severe weight gain, hepatic steatosis and liver injury which indicate that CAR protects from metabolic disorders in female group. This study is well written and organized, and there are some mistakes need to be modified or clarified as followed:

1.  Is there any explanation for the obvious increasing of COL1A1 which is a key indicator of hepatic stellate cells activation?

2.  There is still some spelling mistakes to be modified.

Author Response

We thank the reviewer for this question. COL1A1 is a gene that codes for a subunit of type I collagen, which is a major component of the extracellular matrix (ECM) in various tissues, including the liver. Hepatic stellate cells (HSCs) are a type of liver cell that plays a crucial role in liver fibrosis. When the liver is injured, HSCs may be activated and contribute to the excessive production of ECM components, including type I collagen, leading to fibrosis, which is the formation of scar tissue in the liver. The elevated COL1A1 expression contributes to the accumulation of type I collagen within the extracellular matrix, resulting in the formation of fibrotic tissue. Within our research, the triggering of Col1a1 expression serves as an indicator of the shift from steatosis to steatohepatitis, marked by inflammation and fibrosis as elaborated in the discussion segment.

The spelling errors have been corrected in the text.

This manuscript is a resubmission of an earlier submission. The following is a list of the peer review reports and author responses from that submission.

Round 1

Reviewer 1 Report

In general, this is a fine investigation of the dimorphic roles of CAR on NAFLD pathogenesis. The experiments are well-designed, and the results are interesting. Nonetheless, some points are raised.

·         Authors stated that “CAR deletion exacerbates HFD-induced hyperglycaemia and hyperinsulinemia”. Nonetheless, the results seem rather contradictory to me in females, and authors should be more cautious on their conclusions on this matter. GTT should a better glucose metabolism in CAR KO. Contradictorily, fast insulin and glucose was higher in this group. Please provide a proper explanation for it.

·         Authors should explain HOW females have Epidydimal WAT, since they do not have this organ. This is not right. In addition, authors should present retroperitoneal and mesenteric WAT weight (and a sum of all of them);

·         Authors should consider on evaluating female and male data altogether. In the transcriptome analysis, heatmaps and PCoA data address both male and female, while the other data encompass pairwise comparisons.

·         The Figure 5 (comparison male vs female) lacks statistical analysis;

·         The functional analysis of the genes is deceiving, as many “background” or “confusing” terms are presented: “HIV and Epstein bar infection” and “Huntington, Parkinson diseases”.

·         Authors should refer to the female CAR KO transcriptome comparison with ERa KO to IN SILICO analysis in the methodology. It is not also clear if the GSE95283 data set encompasses a similar HF protocol (i.e., how many months under HF, which was the background mouse strain).

·         Figure 6A, showing a p value is unnecessary and should be replaced by a text citation.

·         Line 39 - “Hepato-Carcinoma“ is not the right nomination for “Hepatocellular Carcinoma”

·         Line 71 - “n=10 mice per group to have a good statistical power”. “Good” is not scientific. “In order to achieve statistical robustness” (or something)

·         The reproduction of NAS parameters in the Table 1 is unnecessary. Two or three lines of brief explanation are enough.

·         Table 3 should be presented as Supplementary material;

·         All data were non parametric? If data was parametric, Mann-Whitney should not be applied, but Student t test.

·         In the same matter, if data were all parametric, mean +/- standard error of the mean is not the proper way to show all data in the manuscript. Non parametric data should be presented as box plots.

·         Regarding the presentation of the results, authors do not need to present mean +SEM in BOTH the text and figures. It is redundant.

·         The meaning of the asterisks is not described in the footnotes. In fact, data on replicates and a brief explanation of the protocol in each footnote would be interesting for the readers;

·         Excessive methodological details in footnotes are also unnecessary. Example: “Plasmatic levels of Alanine Aminotransferase (ALT) and Aspartate Aminotransferase (AST) were assayed using a biochemical analyzer”

·         Line 249 – “Finally, no hepatocyte ballooning was observed in any groups.” Ballooning IS NOT a feature observed in western diet-induced NAFLD models in any case… Please remove this sentence. Authors should not misunderstand “ballooning” with “hypertrophy”

·         As authors do not find a difference in fibrosis score, Sirius red representative photomicrographs are deceiving. Please replace them. In addition to the score, authors should perform the morphometry of collagen fibers (area in % occupied by Sirius red).

·         Lines 354-357- This is a paragraph of manuscript preparation template. Please be more cautious on preparing a manuscript.

Author Response

We thank the reviewer for his comments.

Reviewer 2 Report

In this manuscript, the authors have studied the protective role of CAR against  diet-induced NAFLD and associated metabolic disorders in mice. The authors compared the role of CAR in male and female mice. Overall, this study is of good novelty, and well organized. However, some issues should be clarified.

(1) Why a choline/folic deficient diet, and male LXRαβ-/- knock-out mice were chosen to study in line 141-142?

(2) Please present the results (body weight, Subcutaneous WAT, Epidydimal WAT, AUC, Fasted glycaemia and insulinemia levels, et al.) of female ERα-/- knock-out mice fed with HFD in this manuscript.

(3) Why the Epidydimal WAT decreased in CAR-/- male mice compared to WT male mice? Please discuss it in the discussion part.

(4) Why CAR-/- female mice showed better glucose tolerance than WT mice? Please discuss it in the discussion part.

(5) CAR activation in MCD diet-fed NASH mice (methionine-choline deficient diet) was reported to worsen hepatic fibrosis, while herein CAR showed protective effect against diet-induced NAFLD, please discuss why the results were contradictory?

(6) Although the authors have done lots of experiments (hepatic steatosis, hepatic transcriptome, and hypothalamic astrogliosis analysis), while the logic relationship should be considered in this manuscript, please rewritten the discussion part to connect your results better.

Author Response

We thank the reviewer for his comments. 

Reviewer 3 Report

In this manuscript, Oliviero and the colleagues reported the effects of CAR on diet-induced mouse NAFLD using CAR deficient mice (male and female). They found that CAR deletion in female mice were associated with worsening of NAFLD and astrogliosis in the hypothalamus. The effects of CAR on NAFLD has been already reported by another studies (ref. 13, 14, 15, 16). In addition, one think thata the results of this study, especially transcriptome analyses, would be confusing for readers.

Major comments

1.     In this study, the authors performed transcriptome analyses in mouse livers. They demonstrated the outline of these analyses. However, one could not understand the results. The authors should indicate which factors in which pathways are important for the better understanding of authors. Why did liver cholesterol increase in male mice, and why did liver cholesterol and TG increase in female mice? To investigate precise mechanism, one think the authors should perform single cell analyses (e.g. hepatocyte, hepatic stellate cell, Kupffer cell, endothelial cells…..). The effects of CAR on NAFLD progression were already reported by other researchers, the authors should demonstrate novel findings.

2.     How about the lipid profiles in blood? The data of blood lipid profile would help readers understanding. In addition, the authors should discuss about the lipid metabolism according to blood/liver lipid profiles and transcriptome analyses.

3.     The authors demonstrate that CAR deletion exacerbated hepatic steatosis and injury, but did not affect liver fibrosis. Usually, exacerbated liver injury leads to enhanced liver fibrosis. The authors only demonstrate histological liver fibrosis staining figure and Pdgfr1β gene expression. The authors should demonstrate other fibrosis gene expression (e.g. Col1a1, Acta2, Tgf-b1…). In addition, they should perform in vivo analyses for longer period which would induce fibrotic changes more precisely. The authors should perform in vitro study using hepatic stellate cells collected from each group mouse.

4.     In this study, the authors demonstrate the histological findings in hypothalamus. It seems abrupt  that these data about hypothalamus appear in this manuscript. Why the authors perform these analyses in this study? One could not understand the data about hypothalamus had any effects on liver histology. The authors should describe the relationships between the results of liver and hypothalamus.  

Minor comments

1.     Please demonstrate the composition of HFD precisely in supplementary tables.

2.     The authors described the criteria of liver histological assessment in Table 1. The authors should demonstrate the representative histology that they diagnosed in this study in each histological scores (supplementary figures).

3.     The descriptions in statistical analysis would not be precise. Please describe more precisely.

4.     The abbreviation in Supplementary Figure 1B would be wrong. In stead of “HFD”, “CAR-/-” would be correct.

Author Response

We thank the reviewer for his comments. 

Reviewer 4 Report

The authors have investigated that constitutive androstane nuclear receptor may play a role in the dimorphic pathogenesis of non-alcoholic fatty liver disease. The findings suggest that there may be sex-specific differences in the development of NAFLD, with male mice being more susceptible to HFD-induced weight gain and hepatic steatosis than female mice. The identification of this target may lead to the development of new treatments that are more effective than current therapies for NAFLD/NASH, which have limited efficacy and significant side effects.

There are several potential shortcomings of the above study that should be considered:

Abbreviation should not be used in the title. For this, the title should be revised.

The study only examined the role of CAR in the pathogenesis of NAFLD/NASH and did not investigate the potential involvement of other factors. For example, obesity is a major risk factor for the development of NAFLD, and the study did not examine the impact of obesity on the findings.

Author Response

We thank the reviewer for his comments. 

Round 2

Reviewer 1 Report

Authors ignored most of the suggestions made in the manuscript. The paper should be rejected, as the paper bears some conceptual errors. Some critical examples below:

1) There is no epididymal adipose tissue in females. In their response, authors clarified that they referred to "gonadal" adipose tissue, but no change was made in the manuscript.

2) The sampling of adipose tissue seemed inadequate, as "only perigonadal and subcutneous" AT was collected (according to the author's response. The study lacks mesenteric and retroperitoneal AT weights, and an adiposity index (AI%) measurement.

3) Authors insist on a lenghty methodology description (e.g, Table 2, that is unecessary). 

4) There is no balooning in diet induced mouse models. Only in CCl4+HF (or WD). This is a conceptual error.

5) Authors ignored a simple suggestion to perform the morphometric analysis of collagen fibers in Sirius red stained sections

Reviewer 3 Report

The authors did not adequately response to my major comments. I would like to ask them the same comments to original manuscript.